# Diuretic Effects of Sodium Glucose Cotransporter 2 Inhibitors and Their Influence on the Renin-Angiotensin System

**DOI:** 10.3390/ijms20030629

**Published:** 2019-02-01

**Authors:** Tuba M. Ansary, Daisuke Nakano, Akira Nishiyama

**Affiliations:** Department of Pharmacology, Faculty of Medicine, Kagawa University, 1750-1 Ikenobe, Miki-cho, Kita-gun, Kagawa 761-0793, Japan; tubapsy@med.kagawa-u.ac.jp (T.M.A.); dnakano@med.kagawa-u.ac.jp (D.N.)

**Keywords:** renin-angiotensin system (RAS), sodium glucose cotransporter 2 (SGLT2) inhibitor, diuretic effect, natriuresis, type 2 diabetes

## Abstract

The renin-angiotensin system (RAS) plays an important role in regulating body fluids and blood pressure. However, inappropriate activation of the RAS contributes to the pathogenesis of cardiovascular and renal diseases. Recently, sodium glucose cotransporter 2 (SGLT2) inhibitors have been used as anti-diabetic agents. SGLT2 inhibitors induce glycosuria and improve hyperglycemia by inhibiting urinary reabsorption of glucose. However, in the early stages of treatment, these inhibitors frequently cause polyuria and natriuresis, which potentially activate the RAS. Nevertheless, the effects of SGLT2 inhibitors on RAS activity are not straightforward. Available data indicate that treatment with SGLT2 inhibitors transiently activates the systemic RAS in type 2 diabetic patients, but not the intrarenal RAS. In this review article, we summarize current evidence of the diuretic effects of SGLT2 inhibitors and their influence on RAS activity.

## 1. Introduction

Sodium glucose cotransporter 2 (SGLT2) inhibitors are a relatively new class of antidiabetic drugs. Four SGLT2 inhibitors—canagliflozin, dapagliflozin, empagliflozin, and ertugliflozin—are currently approved by the US Food and Drug Administration (FDA) as glucose-lowering drugs. Two more SGLT2 inhibitors, sotagliflozin and bexagliflozin, are currently in phase III clinical trials for type 2 diabetes. In Japan, three more SGLT2 inhibitors—ipragliflozin, luseogliflozin, and tofogliflozin—have been approved [1]. A few different properties among the SGLT2 inhibitors have been reported. For example, in vitro studies have shown that canagliflozin is less selective between SGLT2 and SGLT1 than other SGLT2 inhibitors [2]. Pharmacokinetics studies have also indicated that ipragliflozin and luseogliflozin show rapid drug distribution in the kidney [3].

In healthy individuals, filtered glucose is fully reabsorbed at the proximal tubules and no glucose is detected in the urine. SGLT2, a low-affinity and high-capacity glucose transporter, is located in the convoluted proximal tubule and responsible for reabsorbing around 90% of filtered glucose in the kidney [4]. In type 2 diabetes, the expression and activity of SGLT2 are significantly increased, which may lead to a further increase in glucose reabsorption and thus contribute to hyperglycemia [5]. Treatment with SGLT2 inhibitors improves hyperglycemia by inhibiting reabsorption of filtered glucose, thereby increasing glycosuria [6,7]. SGLT2 inhibitors reduce the capacity for renal glucose reabsorption by 30–50% [8], but 36–44% of glucose reabsorption is still maintained under SGLT2 deficiency [9,10]. Treatment with SGLT2 inhibitors provides an insulin-independent reduction in hemoglobin A1c levels with potential additional benefits, such as body weight loss, uricosuria, natriuresis, and osmotic diuresis [11,12].

The risk of cardiovascular events and renal diseases is greater in diabetic patients, leading to increased risk of mortality [13]. Recently, the EMPA-REG OUTCOME study and CANVAS program have shown that the selective SGLT2 inhibitors, empagliflozin and canagliflozin, significantly decrease the risk of cardiovascular death or hospitalized heart failure in type 2 diabetic patients who suffer a high risk of cardiovascular diseases [14,15]. The mechanisms behind this beneficial outcome are not well understood, although both empagliflozin and canagliflozin result in reduced blood pressure. Notably, the reduced risk of heart failure occurred during the early phase of the follow-up period, suggesting the possible role of hemodynamic changes induced by SGLT2 inhibitors. Indeed, clinical studies have shown that SGLT2 inhibitors initially cause natriuresis after treatment begins [16,17,18,19]. SGLT2 inhibitors do not have long-term natriuretic effects in type 2 diabetes patients, and the literature on the mechanisms of the transient natriuretic effects of SGLT2 inhibitors is limited. It has been suggested that there is a compensatory mechanism. For example, several sodium transporters may be activated to increase sodium uptake in the tubule in response to the mild natriuresis caused by SGLT2 inhibitors [20]. It is also possible that the diuresis, natriuresis, and associated body fluid loss induced by SGLT2 inhibitors activate the renin-angiotensin system (RAS). Therefore, this review aims to discuss the diuretic effects of SGLT2 inhibitors and their influence on the activity of the RAS. 

## 2. Diuretic Effects of SGLT2 Inhibitors

### 2.1. Changes in Urine Volume and Urinary Sodium Excretion

The diuretic actions of SGLT2 inhibitors presumably play an important role in cardioprotection, as shown in the EMPA-REG OUTCOME study and the CANVAS program. SGLT2 inhibitors have acutely caused an increase in urinary sodium excretion in non-diabetic rats [21] and in diabetic rats [22,23]. In type 2 diabetic patients, increased urinary sodium excretion has been observed during the early phase of treatment with canagliflozin [16,18,19] and empagliflozin [24]. Antihypertensive effects found in the EMPA-REG OUTCOME study and the CANVAS program are probably due to natriuresis induced by the SGLT2 inhibitors [14,15]. Notably, dapagliflozin has been shown to reduce plasma volume in a similar way to thiazide diuretics, but dapagliflozin has a more enduring diuretic effect than other diuretics [25]. The plasma volume reduction is accompanied by an increase in hematocrit, which has been observed in patients treated with SGLT2 inhibitors [8,26]. Likewise, empagliflozin-treated patients have shown a hematocrit of approximately 5% higher than the placebo-treated patients [14]. Mathematical models have indicated that SGLT2 inhibitors can result in interstitial fluid clearance without changing the intravascular volume by osmotic diuresis [27]. 

Over the last decade, sodium accumulation in tissues has received growing attention as a marker of volume-expanded states [28]. Sodium concentrations in the skin and muscles are reportedly positively correlated with the risk of cardiovascular diseases and blood pressure in chronic kidney disease [29,30]. Interestingly, chronic treatment with dapagliflozin significantly reduces sodium concentrations in the skin of type 2 diabetic patients [31]. These data support the hypothesis that SGLT2 inhibitors decrease the sodium concentration in interstitial fluid and thus reduce the risk of cardiovascular diseases. 

Clinical studies have shown that canagliflozin increases urine volume in type 2 diabetic patients during the first few days of treatment [18,32,33,34,35,36]. In Japanese type 2 diabetic patients, empagliflozin increases urine volume initially [37]. Similar observations have also been found in type 2 diabetic patients treated with dapagliflozin [38]. Collectively, data consistently indicate that treatment with SGLT2 inhibitors results in natriuresis that is associated with glycosuria. However, these diuretic effects appear to be transient based on the available data in patients with type 2 diabetes. The effects of SGLT2 inhibitors on urine volume and urinary sodium excretion are summarized in Table 1.

### 2.2. Changes in Tubular Functions

It has been indicated that transport of water and sodium is activated in diabetes to compensate loss of water and sodium [55]. In type 1 diabetic patients, expression of renal sodium and water channels, such as sodium-hydrogen exchanger (NHE3), Na–Cl cotransporter (NCC), epithelial sodium channel (ENaC), Na–K–2Cl cotransporter (NKCC2), aquaporin 2, and urea transporters is significantly increased [56,57,58]. Under hyperglycemic conditions, human exfoliated proximal tubular epithelial cells isolated from type 2 diabetic patients have a threefold increase of glucose uptake along with increased expression and activity of SGLT2 [7]. The expression of distal nephron sodium transporter proteins, such as NKCC2, NCC and ENaC, is also increased in type 2 diabetes, resulting in increased sodium reabsorption [59,60]. Studies have shown that total sodium reabsorption by SGLT2 is significantly increased in diabetes [61].

Microperfusion studies have also shown that SGLT2 functionally interact and colocalize with NHE3 in the proximal tubule and that inhibition of SGLT2 with phlorizin can inhibit NHE3-dependent bicarbonate reabsorption [62]. Moreover, phosphorylation of NHE3, which associates with its reduced activity, is increased in type 1 diabetic Akita mice treated with empagliflozin [63,64]. Conversely, another study has shown that treatment with luseogliflozin increases NHE3 and NCC protein expression in the renal cortical tissues of spontaneously hypertensive rats [20], probably owing to the compensatory increase in solute delivery in the tubules by the SGLT2 inhibitor. Thus, the effects of SGLT2 inhibitors on the function of sodium transporters appear to be controversial. 

Dapagliflozin has been shown to reduce proteinuria in an experimental model of non-diabetic proteinuric nephropathy without affecting glomerular filtration rate (GFR) [65]. Studies in db/db mice have also shown that hyperglycemia increases kidney volume without enlarging glomeruli and that treatment with SGLT2 inhibitors causes glomerular hypertrophy, glomeruli redistribution, and reduced kidney volume, with no effects on GFR [66]. Similarly, acute administration of luseogliflozin has not changed GFR in non-diabetic rats [21]. However, empagliflozin has resulted in a significant decrease in GFR in Akita mice [67]. Collectively, while SGLT2 inhibitors frequently decrease GFR in diabetic patients, this effect is not consistently observed in diabetic and non-diabetic animals. However, the active site for the SGLT2 inhibitors and molecular mechanisms responsible for the diuresis and natriuresis induced has not been determined yet.

## 3. Effects of SGLT2 Inhibitors on RAS Activity

Inappropriate activation of the RAS is a critical factor in the pathogenesis of cardiovascular and renal diseases associated with diabetes [13]. In response to stimuli such as reduced renal perfusion pressure and sodium load at the distal tubules, renin release from the juxtaglomerular cells is significantly increased [68]. SGLT2 inhibitors can cause natriuresis, at least in the early phase of treatment, resulting in systemic RAS activation. Furthermore, the intrarenal RAS is sometimes activated to compensate for the sodium and water loss caused by SGLT2 inhibitors [69]. To monitor the RAS activity, several biomarkers have been used, including plasma renin activity (PRA), plasma aldosterone concentration, urinary angiotensinogen (AGT) and angiotensin II (Ang II) levels [70,71,72,73]. Effects of SGLT2 inhibitors on RAS parameters are summarized in Table 2 and discussed in detail later.

### 3.1. Systemic RAS

In patients with essential hypertension, treatment with diuretics often elevates PRA, probably because of body fluid loss [78,79]. Similarly, SGLT2 inhibitor-induced osmotic diuresis, natriuresis, and the associated reductions in extracellular volume and blood pressure might activate the systemic RAS in diabetes [25,46,50,80]. Type 2 diabetic db/db mice have shown higher PRA than the non-diabetic counterparts, but treatment with empagliflozin has not changed the increased PRA [46]. In Otsuka Long-Evans Tokushima fatty (OLETF) rats, a type 2 diabetic model, PRA and serum aldosterone levels are significantly increased and remain unchanged upon chronic treatment with dapagliflozin [74]. Similarly, the levels of systemic RAS components do not change significantly after 10 weeks of SGLT2 inhibitor treatment in a chronic kidney disease rat model [69]. In clinical studies, dapagliflozin has increased PRA and serum aldosterone after 12 weeks of treatment in type 2 diabetic patients [25]. Conversely, PRA did not change significantly after chronic treatment of type 2 diabetic patients with SGLT2 inhibitors in another study [76]. Increases in systemic RAS parameters can be explained by the compensatory mechanism in response to volume reduction by SGLT2 inhibitors. However, available clinical data have shown that the plasma aldosterone level did not significantly change by treatment with an SGLT2 inhibitor [74]. This may be because aldosterone production is stimulated not only by Ang II, but also by other factors, such as adrenocorticotropic hormone and potassium. Furthermore, the aldosterone level is affected by sampling conditions because it is driven by the circadian rhythm [81]. Moreover, as mentioned before, the diuretic action of SGLT2 inhibitors is usually transient and is no longer observed a few days after treatment [16,17,18,19,82,83]. 

Furthermore, SGLT2 inhibitors improve the circadian rhythm of sympathetic nerve activity [84,85], while their effects on sympathetic nerve activity may also influence the activity of the systemic RAS.

### 3.2. Intrarenal RAS

Studies have indicated that the liver is the source of AGT in both plasma and kidneys [86]. However, it has also been shown that AGT is locally expressed in proximal tubules [87]. As shown in Figure 1, treatment with an SGLT2 inhibitor may affect intrarenal AGT production via changes in glucose levels. However, it is unclear whether SGLT2 inhibitors influence systemic AGT filtration at the glomerulus. It has been shown that AGT plays a predominant role in the regulation of intrarenal RAS [68,72]. Administration of high glucose (15 mM) has significantly increased AGT mRNA levels in cultured human proximal tubular cells [88]. Therefore, a reduction in blood sugar levels with SGLT2 inhibitors may decrease AGT production in the early proximal tubule. Conversely, SGLT2 inhibition can increase glucose delivery to the distal proximal tubule, and, therefore, AGT production may be stimulated. Different effects of SGLT2 inhibitors on AGT production between early and distal proximal tubules may explain the inconsistent data regarding the responses of intrarenal RAS to SGLT2 inhibitors. Type 2 diabetic OLETF rats have shown augmented intrarenal RAS [89], and treatment with dapagliflozin significantly lowered the urinary Ang II and AGT levels [74]. In patients with type 2 diabetes, treatment with SGLT2 inhibitors also tended to decrease urinary AGT excretion, although these changes were not statistically significant [75]. In contrast, in patients with type 1 diabetes, treatment with empagliflozin significantly increased the urinary AGT/creatinine ratio [50]. Importantly, several studies have indicated a higher urinary AGT/creatinine ratio in patients with type 2 diabetes than in patients with type 1 diabetes [90,91]. Thus, the basal activity of intrarenal RAS may influence the SGLT2 inhibitor’s effect on it. Another important point is that diabetic patients are frequently medicated with RAS inhibitors, such as angiotensin-converting enzyme inhibitors and angiotensin receptor blockers [14], which may affect the changes in intrarenal RAS in type 2 diabetes. Nevertheless, we cannot find any evidence that SGLT2 inhibitors actually activate the intrarenal RAS in type 2 diabetes. The possible mechanisms by which SGLT2 inhibitors influence the systemic and renal RAS are illustrated in Figure 1.

## 4. Conclusions

The findings that the beneficial effects of SGLT2 inhibitors on cardiovascular outcomes are observed in the early phase of treatment support the direct diuretic and hemodynamic effects of SGLT2 inhibitors. The transient diuretic effects potentially lead to systemic RAS activation. However, the association between SGLT2 inhibitors and systemic RAS activation is not straightforward. Furthermore, data have indicated that chronic treatment with SGLT2 inhibitors does not activate the intrarenal RAS in type 2 diabetic patients. Future work is thus needed to develop stable biomarkers for both the systemic and intrarenal RAS. To contribute to this unmet need, we have recently developed new assay systems for measuring stable PRA and intrarenal renin activity by calculating the ratio of (total AGT−intact AGT)/intact AGT in plasma and urine respectively [68,69,92].

## Figures and Tables

**Figure 1 ijms-20-00629-f001:**
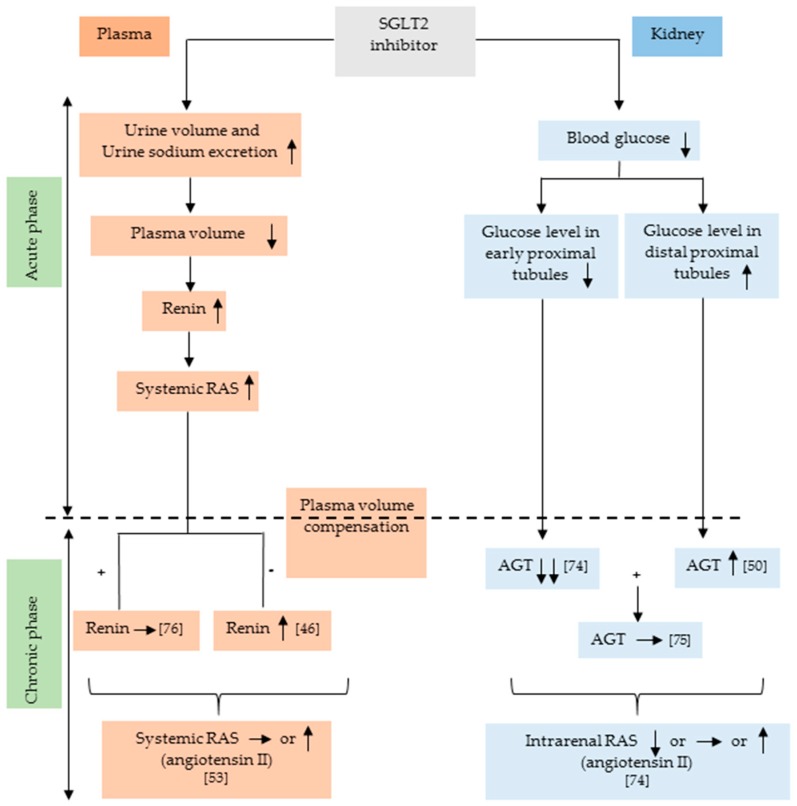
Possible mechanisms by which SGLT2 inhibitors influence the systemic and intrarenal RAS. SGLT2 inhibitors transiently increase plasma renin activity acutely through osmotic diuresis. Meanwhile, SGLT2 inhibitors decrease renal AGT expression by reducing glucose levels in the kidney. However, SGLT2 inhibitors can increase the glucose load in distal proximal tubule and that might increase the AGT production. SGLT2, sodium-glucose cotransporter 2; RAS, renin-angiotensin system; AGT, angiotensinogen. ↑, increase; ↓, decrease; →, no change; +, in case of plasma volume compensation; −, in case of no plasma volume compensation.

**Table 1 ijms-20-00629-t001:** SGLT2 inhibitor-induced changes in urine volume and urinary sodium excretion.

	Subjects	Observation Period	Food Restriction, etc.	Urinary Sodium Excretion	Urine Volume	SGLT2 Inhibitor	Reference
**Animal Experiments**	ZDF rats	24 h	No	N/A	Increased by ~1.82 fold	dapagliflozin	[39]
	SD rats				Increased by ~5.0 fold		
	Dogs	24 h	No	Increased by ~1.50 fold	Increased by ~3.7 fold	bexagliflozin or EGT1442	[40]
	ZDF rats	24 h	No	Increased by ~1.80 fold	Increased by ~1.82 fold	luseogliflozin	[41]
	db/db mice	4 weeks	No	N/A	Increased by ~1.27 fold	tofogliflozin	[42]
		8 weeks			No change		
	Streptozotocin-nicotinamide induced type 2 diabetic mice	24 h	High fat diet	N/A	Increased by ~4.6 fold	ipragliflozin	[43]
	db/db mice	Day 1	No	Increased by ~1.40 fold	Increased by ~1.67 fold	empagliflozin	[44]
		7 days		No change	Increased by ~1.33 fold		
	Streptozotocin induced type 1 diabetic rats	70 min	No	Increased by ~4.0 fold	Increased by ~2.28 fold	phlorizin	[45]
	SHRcp rats	5 weeks	No	Increased by ~1.30 fold	Increased by ~4.0 fold	luseogliflozin	[20]
	OLETF rats	12 h	0.5% NaCl diet	Increased by ~1.30 fold	N/A	empagliflozin	[23]
		5 weeks		N/A			
	db/db mice	10 weeks	No	N/A	Increased by ~1.06 fold	empagliflozin	[46]
	Streptozotocin induced type 1 diabetic mice	3 weeks	No	N/A	Increased by ~1.17 fold	ipragliflozin	[47]
	C57BL/6J mice	16 weeks	High fat diet	N/A	Increased by ~4.0 fold	empagliflozin	[48]
	SD rats	120 min	No	Increased by ~7.0 fold	Increased by ~1.90 fold	luseogliflozin	[21]
**Clinical Studies**	Type 2 diabetes	12 weeks	Standard diet	No change	Increased by ~.26 fold	dapagliflozin	[38]
	Type 2 diabetes	12 weeks	Standard diet	N/A	Increased by ~1.25 fold	dapagliflozin	[49]
	Type 1 diabetes with hyperfiltration	8 weeks	High sodium (>140 mmol/d) and moderate protein (<1.5 g/kg/d) diet for 14 days	Increased by ~1.10 fold	Increased by ~1.56 fold	empagliflozin	[50]
	Type 2 diabetes	12 weeks	Sodium restricted diet (∼200 mmol/day)	Increased by ~1.22 fold	Increased by ~1.49 fold	canagliflozin	[18]
	Type 2 diabetes	Day 1	Isocaloric diet	N/A	Increased by ~1.14 fold	canagliflozin	[51]
		2 weeks			No change		
	Type 2 diabetes	7 days	Standardized meal of approximately 600 kcal	N/A	Increased by ~1.19 fold	luseogliflozin	[52]
	Type 2 diabetes	4 days	No	Increased by ~1.28 fold	Increased by ~1.6 fold	ipragliflozin	[53]
	Type 2 diabetes	24 h	Standard diet	No change	Increased by ~1.27 fold	canagliflozin	[19]
		18 days			No change		
	Type 2 diabetes	Day 1	No	Increased by ~1.33 fold	Increased by ~3.71 fold	canagliflozin	[16]
		Day 5		No change	Increased by ~1.03 fold		
	Type 2 diabetes	6 months	Standard diet	Increased by ~1.40 fold	Increased by ~1.86 fold	ipragliflozin, dapagliflozin, tofogliflozin, luseogliflozin	[54]

SGLT2, sodium glucose cotransporter 2; ZDF, zucker diabetic fatty; SD, sprague dawley; SHRcp, metabolic syndrome SHR/NDmcr-cp (+/+); OLETF, otsuka long-evans tokushima fatty; N/A, not available.

**Table 2 ijms-20-00629-t002:** Relationship between SGLT2 inhibitor-induced changes in urine volume and urinary sodium excretion and RAS parameters.

	Subjects	Observation Period	RAS Parameters	Urinary Sodium Excretion	Urine Volume	SGLT2 Inhibitor	Reference
**Animal Experiments**	db/db mice	10 weeks	PRA increased by ~1.5 fold	N/A	Increased by ~1.06 fold	empagliflozin	[46]
	5/6 Nx SD rats	10 weeks	No change	N/A	N/A	luseogliflozin	[69]
	OLETF rats	12 weeks	PRA no change	N/A	N/A	dapagliflozin	[74]
			Urinary Ang II decreased by ~30 fold				
			Urinary AGT decreased by ~5 fold				
			Plasma aldosterone no change				
**Clinical Studies**	Type 2 diabetes	Day 1	PRA no change	Increased by ~1.33 fold	Increased by ~3.71 fold	canagliflozin	[16]
			Plasma aldosterone no change				
		Day 5	PRA increased by ~2 fold	No change	Increased by ~1.03 fold		
			Plasma aldosterone no change				
	Type 1 diabetes with hyperfiltration	8 weeks	PRA increased by ~1.11 fold, but this change was not significant	Increased by ~1.10 fold	Increased by ~1.56 fold	empagliflozin	[50]
			Plasma Ang II increased by ~1.56 fold				
			Plasma aldosterone increased by ~1.72 fold				
	Type 2 diabetes	4 days	PRA increased by ~1.0 fold, but this change was not significant	Increased by ~1.28 fold	Increased by ~1.6 fold	ipragliflozin	[53]
			Plasma Ang II no change				
			Plasma aldosterone no change				
	Type 2 diabetes	1 month	total urinary AGT/creatinine ratio no changed	N/A	N/A	canagliflozin, ipragliflozin, dapagliflozin, tofogliflozin, luseogliflozin	[75]
	Type 2 diabetes	1 month	PRA increased by ~3.0 fold	N/A	N/A	tofogliflozin, empagliflozin, canagliflozin	[76]
			Plasma aldosterone no change				
		6 months	PRA no change				
			Plasma aldosterone no change				
	Type 2 diabetes	24 weeks	PRA increased by ~1.59 fold, but this change was not significant	N/A	N/A	Ipragliflozin	[77]
			Plasma aldosterone increased by ~1.27 fold				

SGLT2, sodium glucose cotransporter 2; RAS, renin angiotensin system; PRA, plasma renin activity; Nx, nephrectomy; SD, sprague dawley; OLETF, otsuka long-evans tokushima fatty; Ang II, angiotensin II; AGT, angiotensinogen.

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
