# Peer review of "Diuretic Effects of Sodium Glucose Cotransporter 2 Inhibitors and Their Influence on the Renin-Angiotensin System"

_ijms, 2019, doi:10.3390/ijms20030629_

Reviewer 1 Report

    The author reviewed diuresis and natriuresis caused by SGLT2 inhibitors in T2DM. Furthermore, they discussed previous findings regarding regulation of RAS levels by SGLT2 inhibitions. The manuscript adequately summarizes past research findings and provides current scientific issues which must be addressed in the research field. Thus, the article will be interesting and helpful for potential readers. However, there are further minor considerations that could improve this article.

1) Because SGLT2 is a sodium-glucose co-transporter, SGLT2 inhibition reduces reabsorption of sodium as well as glucose at renal proximal tubules. However, sodium excretion is augmented by the inhibitors at early phage of treatments as summarized in the manuscript. Thus, describing possible mechanisms underlying natriuresis only in the early phase of SGLT2 treatment will be helpful.

2) Although the tables include new SGLT2 inhibitors including luseogliflozin, there is no explanation about classifications and differences among these SGLT2 inhibitors in the text. Accordingly, it would be better to add brief descriptions explaining differences among the inhibitors to the Introduction section.

3) It would be interesting to propose additional possible reasons for the inconsistent findings in intrarenal RAS regulations by SGLT2 inhibitors. For example, SGLT2 inhibition may attenuate angiotensinogen augmentation in the early segment of proximal tubules in DM. On the other hand, the blockage of glucose reabsorption via SGLT2 will, at least transiently, increase glucose levels in proximal tubular fluid at the late segments which can stimulate AGT expression at these segments. Furthermore, hyperglycemia-induced AGT regulating factors including ROS, advanced glycation end products and inflammatory cytokine levels can show varying temporal changes during treatment with the inhibitors.

Author Response

Responses to Reviewer 1’s Comments

General Comments: 

The author reviewed diuresis and natriuresis caused by SGLT2 inhibitors in T2DM. Furthermore, they discussed previous findings regarding regulation of RAS levels by SGLT2 inhibitions. The manuscript adequately summarizes past research findings and provides current scientific issues which must be addressed in the research field. Thus, the article will be interesting and helpful for potential readers. However, there are further minor considerations that could improve this article.

We thank the reviewer for the thoughtful review and crucial comments, which have helped guide the revision of the manuscript. Our replies are as follows:

Point 1: Because SGLT2 is a sodium glucose cotransporter, SGLT2 inhibition reduces reabsorption of sodium as well as glucose at renal proximal tubules. However, sodium excretion is augmented by the inhibitors at early phage of treatments as summarized in the manuscript. Thus, describing possible mechanisms underlying natriuresis only in the early phase of SGLT2 treatment will be helpful.

Response 1: We appreciate the reviewer’s comment regarding the possible mechanisms underlying natriuresis only in the early phase of SGLT2 treatment. As pointed out, SGLT2 inhibitors increase urinary sodium excretion in the early stage of treatment. However, increased urinary sodium excretion returns to the baseline levels within a few days. The reason behind this may be the activation of compensatory mechanisms. Although the detailed compensatory mechanism is not clear, we discussed the possible mechanisms of transient natriuresis during SGLT2 treatment in the revised manuscript (page 2, lines 44–48)

“Clinical studies have shown that SGLT2 inhibitors initially cause natriuresis after treatment begins [16-19]. SGLT2 inhibitors do not have long-term natriuretic effects in type 2 diabetes patients, and the literature on the mechanisms of the transient natriuretic effects of SGLT2 inhibitors is limited. It has been suggested that there is a compensatory mechanism; for example, several sodium transporters may be activated to increase sodium uptake in the tubule in response to the mild natriuresis caused by SGLT2 inhibitors [20].”

Point 2: Although the tables include new SGLT2 inhibitors including luseogliflozin, there is no explanation about classifications and differences among these SGLT2 inhibitors in the text. Accordingly, it would be better to add brief descriptions explaining differences among the inhibitors to the Introduction section.

Response 2: We apologize for our insufficient explanation of the classification of SGLT2 inhibitors. As suggested, we added detailed information in the revised manuscript (page 2, lines 16–23).

“Sodium glucose cotransporter 2 (SGLT2) inhibitors are a relatively new class of antidiabetic drugs. Four SGLT2 inhibitors; canagliflozin, dapagliflozin, empagliflozin and ertugliflozin are currently approved by the FDA as glucose-lowering drugs. Two more SGLT2 inhibitors, sotagliflozin and bexagliflozin are currently in phase III clinical trials for type 2 diabetes. In Japan, three moreSGLT2 inhibitors; ipragliflozin, luseogliflozin and tofogliflozin have been approved [1]. A few different properties among the SGLT2 inhibitors have been reported; for example, in vitro studies have shown that canagliflozin is less selective between SGLT2 and SGLT1 than other SGLT2 inhibitors [2]. Pharmacokinetics studies have also indicated that ipragliflozin and luseogliflozin showed rapid drug distribution in the kidney [3].”

Point 3: It would be interesting to propose additional possible reasons for the inconsistent findings in intrarenal RAS regulations by SGLT2 inhibitors. For example, SGLT2 inhibition may attenuate angiotensinogen augmentation in the early segment of proximal tubules in DM. On the other hand, the blockage of glucose reabsorption via SGLT2 will, at least transiently, increase glucose levels in proximal tubular fluid at the late segments which can stimulate AGT expression at these segments. Furthermore, hyperglycemia induced AGT regulating factors including ROS, advanced glycation end products and inflammatory cytokine levels can show varying temporal changes during treatment with the inhibitors. 

Response 3:We thank the reviewer for these excellent comments regarding different regulation of intrarenal AGT production. According to the reviewer’s suggestions, we mentioned these possibilities in detail in our revised manuscript and in Figure 1 (page 9, lines 33–43; page 10, lines 1–11).

“It has been shown that AGT plays a predominant role in the regulation of intrarenal RAS [68,72]. Administration of high glucose (15 mM) has significantly increasedAGTmRNA levels in cultured human proximal tubular cells [88], thus reduction in blood sugar levels with SGLT2 inhibitors may decrease AGT production in the early proximal tubule. On the contrary, SGLT2 inhibition can increase the glucose delivery to the distal proximal tubule, and thus AGT production may be stimulated. Different effects of SGLT2 inhibitors on AGT production between early and distal proximal tubules may explain the inconsistent data regarding the responses of intrarenal RAS to SGLT2 inhibitors. Type 2 diabetic OLETF rats have shown augmented intrarenal RAS [89], and treatment with dapagliflozin significantly lowered the urinary Ang II and AGT levels [74]. In patients with type 2 diabetes, treatment with SGLT2 inhibitors also tended to decrease urinary AGT excretion, although these changes were not statistically significant [75]. In contrast, in patients with type 1 diabetes, treatment with empagliflozin significantly increased urinary AGT/creatinine ratio [50]. Importantly, several studies have indicated a higher urinary AGT/creatinine ratio in patients with type 2 diabetes than in patients with type 1 diabetes [90,91]. Thus, the basal activity of intrarenal RAS may influence the SGLT2 inhibitor effect on it. Another important point is that diabetic patients are frequently medicated with RAS inhibitors, such as angiotensin-converting enzyme inhibitors and angiotensin receptor blockers [14], which may affect the changes in intrarenal RAS in type 2 diabetes. Nevertheless, we cannot find any evidence that SGLT2 inhibitors actually activate the intrarenal RAS in type 2 diabetes. The possible mechanisms by which SGLT2 inhibitors influence the systemic and renal RAS are illustrated in Figure 1.”

Reviewer 2 Report

The summaries of human and animal studies investigating SGLT-2 inhibitors are well organized and comprehensive. The paper is well-written and the topic is very relevant.

Major Issues: 

- The authors focus on Angiotensinogen, PRA and Ang II as when evaluating the impact of SGLT-2 inhibitors on the RAAS. Aldosterone is major component of the Renin-Angiotensin-Aldosterone-System being critically involved in regulating the renal sodium balance. The authors discuss the potential crosstalk between Aldosterone and SGLT-2 inhibition.

- Wherever available, the authors should include information on urinary sodium excretion and urinary volume into Table 2. Given that there is a close connection between sodium balance and the RAS, interpretation of the impact of SGLT-2 inhibitors on the RAAS would be more comprehensive in the context of sodium.

- Figure 1 should be revised. Systemic and local renal compartments should be somehow illustrated to facilitate the reader's understanding of the proposed links. Importantly, details on molecular mechanisms should be provided. E.g.:

- In their final statement, the authors stress their "novel assay" to assess intra-renal renin activity that is based on ratios between intact and total AGT that are compared between plasma and urine. 

In a previous statement, the authors mention that ".... the intrarenal RAS appears to be predominantly regulated by local production of AGT", citing papers by Nishiyama and Kobori. Importantly, papers by Matsusaka et al. showing that the kidney AGT is exclusively derived from the liver were not mentioned at all.

I summary, while clinical aspects are well covered by the authors, mechanistic considerations are incomplete and lack details that would be important for the reader's understanding. More sophisticated illustrations and and mechanistic assessments should be included as a basis for discussing te impact of SGLT-2 inhibitors on the kidney and their crosstalk with the local and the systemic RAAS.

Author Response

Responses to Reviewer 2’s Comments

General Comments: The summaries of human and animal studies investigating SGLT2 inhibitors are well organized and comprehensive. The paper is well written and the topic is very relevant.

We thank the reviewer for the important comments, which have helped guide the revision of our invited review article. Our replies are as follows:

Point 1:The authors focus on Angiotensinogen, PRA and Ang II as when evaluating the impact of SGLT2 inhibitors on the RAAS. Aldosterone is major component of the Renin Angiotensin Aldosterone System being critically involved in regulating the renal sodium balance. The authors discuss the potential crosstalk between Aldosterone and SGLT2 inhibition.

Response 1: We apologize for the missing data regarding changes in aldosterone during SGLT2 inhibitor treatment. In the revised manuscript, we added data (Table 2) and discussionabout SGLT2 inhibition and aldosterone levels (page 9, lines 17–21). 

“Increases in systemic RAS parameters can be explained by the compensatory mechanism in response to volume reduction by SGLT2 inhibitors. However, available clinical data have shown that the plasma aldosterone level did not significantly change by treatment with an SGLT2 inhibitor [74]. This may be because aldosterone production is stimulated not only by Ang II, but also by other factors, such as adrenocorticotropic hormone and potassium.Furthermore, the aldosterone level is affected by sampling conditions because it is driven by the circadian rhythm [81].

Point 2: Wherever available, the authors should include information on urinary sodium excretion and urinary volume into Table 2. Given that there is a close connection between sodium balance and the RAS, interpretation of the impact of SGLT2 inhibitors on the RAAS would be more comprehensive in the context of sodium.

Response 2: In accordance with the reviewer’s suggestion, we added data on urinary sodium excretion and urinary volume in Table 2. We also interpreted the effect of SGLT2 inhibitors on the RAS in the context of sodium in the revised manuscript (page 6, lines 34–36). 

Inappropriate activation of the RAS is a critical factor in the pathogenesis of cardiovascular and renal diseases associated with diabetes [13]. In response to stimuli, such as reduced renal perfusion pressure and sodium load at the distal tubules, renin release from the juxtaglomerular cells is significantly increased [68]. SGLT2 inhibitors can cause natriuresis, at least in the early phase of treatment, resulting in systemic RAS activation. Furthermore, the intrarenal RAS is sometimes activated to compensate for the sodium and water loss caused by SGLT2 inhibitors [69].”

Point 3:Figure 1 should be revised. Systemic and local renal compartments should be somehow illustrated to facilitate the reader's understanding of the proposed links. Importantly, details on molecular mechanisms should be provided. E.g.: 

In their final statement, the authors stress their "novel assay" to assess intrarenal renin activity that is based on ratios between intact and total AGT that are compared between plasma and urine. In a previous statement, the authors mention that ".... the intrarenal RAS appears to be predominantly regulated by local production of AGT", citing papers by Nishiyama and Kobori. Importantly, papers by Matsusaka et al showing that the kidney AGT is exclusively derived from the liver were not mentioned at all.

Response 3:In accordance with the reviewer’s suggestion, we modified Figure 1. We also discussed the local AGT regulation in the kidney (page 9, lines 28–32), as suggested by the reviewer. 

 “Studies have indicated that the liver is the potential source of AGT in both plasma and kidney [86]. However, it  has also been shown that AGT is locally expressed in proximal tubules [87]. As shown in Figure 1, treatment with an SGLT2 inhibitor may affect intrarenal AGT production via changes in glucose levels. However, it is unclear whether SGLT2 inhibitors influence systemic AGT filtration at the glomerulus.”